# The correlation between cell and nucleus size is explained by an eukaryotic cell growth model

**Yufei Wu**[1,5], **Adrian F. Pegoraro**[2], **David A. Weitz**[3], **Paul Janmey**[4], **Sean X. Sun**[1,5,6]*

**1** Department of Mechanical Engineering, Johns Hopkins University, Baltimore, Maryland, United States of America, **2** Department of Physics, University of Ottawa, Ottawa, Canada, **3** Department of Physics, Harvard University, Boston, Massachusetts, United States of America, **4** Department of Cell and Developmental Biology, University of Pennsylvania School of Medicine, Philadelphia, Pennsylvania, United States of America, **5** Institute for NanoBioTechnology, Johns Hopkins University, Baltimore, Maryland, United States of America, **6** Center for Cell Dynamics, Johns Hopkins School of Medicine, Baltimore, Maryland, United States of America

* ssun@jhu.edu

**Data Availability Statement:** All relevant data are within the manuscript and its Supporting information files.

## Abstract

In eukaryotes, the cell volume is observed to be strongly correlated with the nuclear volume. The slope of this correlation depends on the cell type, growth condition, and the physical environment of the cell. We develop a computational model of cell growth and proteome increase, incorporating the kinetics of amino acid import, protein/ribosome synthesis and degradation, and active transport of proteins between the cytoplasm and the nucleoplasm. We also include a simple model of ribosome biogenesis and assembly. Results show that the cell volume is tightly correlated with the nuclear volume, and the cytoplasm-nucleoplasm transport rates strongly influence the cell growth rate as well as the cell/nucleus volume ratio (C/N ratio). Ribosome assembly and the ratio of ribosomal proteins to mature ribosomes also influence the cell volume and the cell growth rate. We find that in order to regulate the cell growth rate and the cell/nucleus volume ratio, the cell must optimally control groups of kinetic and transport parameters together, which could explain the quantitative roles of canonical growth pathways. Finally, although not explicitly demonstrated in this work, we point out that it is possible to construct a detailed proteome distribution using our model and RNAseq data, provided that a quantitative cell division mechanism is known.

## Author summary

We develop computational model of cell proteome increase and cell growth to compute the cell volume to nuclear volume ratio. The model incorporates essential kinetics of protein and ribosome synthesis/degradation, and their transport across the nuclear envelope. The model also incorporates ribosome biogenesis and assembly. The model identifies the most important parameters in determining the cell growth rate and the cell/nucleus volume ratio, and provides a computational starting point to construct the cell proteome based on the RNAseq data.

**Funding:** SXS and YFW are supported by NIH R01GM134542. DAW and AFP are supported by NIH P01HL120839 and Harvard Materials Research Science and Engineering Center Grant DMR-1420570. PJ is supported by NIH R35GM136259. The funders had no role in study design, data collection and analysis, decision to publish, or preparation of the manuscript.

**Competing interests:** The authors have declared that no competing interests exist.

## Introduction

A universal feature of eukaryotic cells is that the nuclear size appears to be correlated with the cell size [1–3]. This correlation has been quantified in yeast and mammalian cells, cells in epithelial tissues [4], and cells growing in different nutrient and physical environments. It occurs in a population of isogenic cells as well as across populations of different cell types (Fig 1a). This correlation also persists in normal as well as cancer cells, although oncogene mutations appear to alter the cell/nucleus volume ratio (i.e., C/N ratio). In growing cells, the cytoplasm is continuously expanding from amino-acid and nutrient import, and also protein synthesis. Proteins and ribosomes are also actively transported in and out of the nucleus. During steady growth, the net nuclear import should balance cellular uptake and synthesis, which ultimately gives rise to the same growth rates of both the nucleus, the cell, and other organelles. This balanced growth idea is well appreciated in the realm of prokaryotic cells [5, 6]. However, for eukaryotes, it is still unclear how the C/N ratio is determined and whether the C/N ratio serves an important functional role. In this paper, we theoretically explore the question of cell-nucleus size correlation using a phenomenological mathematical model. The model predicts that the cell volume is proportional to the nuclear volume, and explicitly shows how various kinetic and transport parameters influence the C/N ratio and the cell growth rate. We find that by incorporating a model of ribosome biogenesis, the model can quantitatively explain cell-nucleus size correlation and the C/N ratio. Moreover, our model suggests that nucleoplasm-cytoplasm transport plays an important role in both the overall cell growth and the C/N ratio. A balanced proteome, (i.e., the right ratio between ribosomal and non-ribosomal protein), is necessary to achieve an optimal growth rate.

For any cell, the overall cell mass increase must come from the net import of extracellular material: water, amino-acids, nutrients, endocytosed and phagocytosed materials, etc. In the cytoplasm, mature ribosome complexes are responsible for protein translation. For eukaryotes, ribosomes undergo a maturation process [7, 8] where new ribosomal proteins are made in the cytoplasm, and then are transported into the nucleoplasm to combine with ribosomal RNA to form large and small subunits. Once matured, the ribosomal subunits are transported out of the nucleus into the cytoplasm and combine into a complete ribosome to make new proteins from mRNA. The transport process across the nuclear envelope is carried out by the RanGTPase cycle, which also transports other proteins across the nuclear envelope [9]. Therefore, the cytoplasmic population of mature ribosomes depends on nuclear-cytoplasmic transport, which in turn determines the protein synthesis rate. Using a simple model, we demonstrate that the rate of nuclear-cytoplasmic transport directly influences the nuclear-cytoplasm ratio, and the cell growth rate. The model is not molecularly detailed, i.e., it does not model individual proteins and their interactions. The model also does not consider the stochastic nature of individual gene expression and protein synthesis [10–13]. Rather, we model the overall population of proteins and ribosomes, and therefore is a coarse-grained cell scale model. Refinement of the modeling framework and connections to RNAseq data and gene regulation models are discussed.

We begin by outlining equations governing the time evolution of the cell proteome number together with ribosome production. We then consider the role of the nucleoplasm-cytoplasm transport in ribosome biogenesis and incorporate this element in the minimal model. We also consider proteolysis or protein degradation. The solution of these equations predicts an exponential increase in assembled proteins, ribosomal proteins, and mature ribosome particle. Moreover, the nucleoplasm and cytoplasm components of proteins and ribosomes all increase at the same rate, and therefore the model predicts that the nuclear volume is proportional to the cytoplasmic volume. We analyze several regimes of the model, including when the

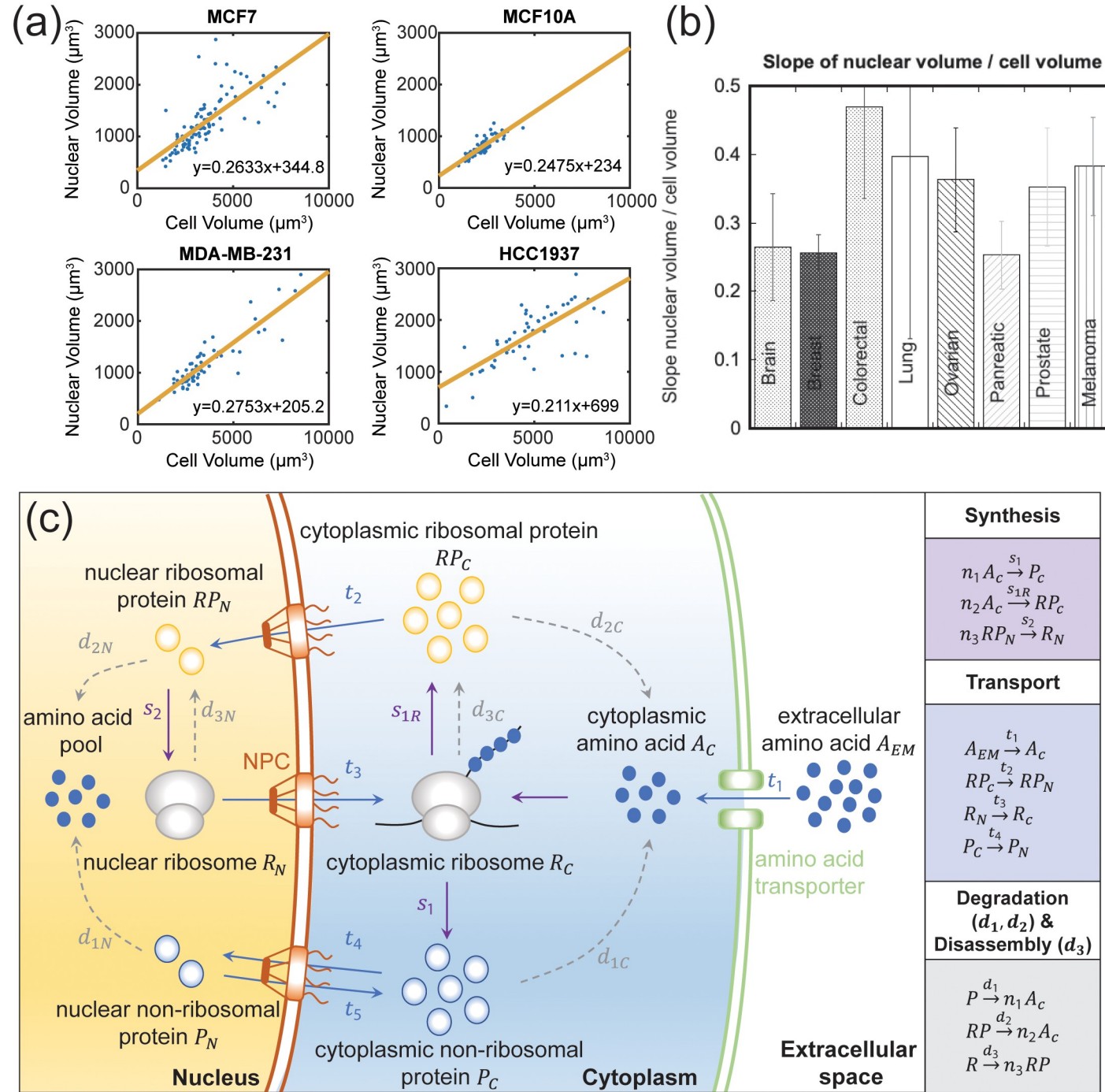

**Fig 1.** (a) Nuclear volume vs. cell volume for several cell lines grown on 500Pa polyacrylamide (PAA) substrates coated with collagen I. Each point is a single cell. The nuclear volume is proportional to the cell volume. (b) Slope of nuclear volume vs. cell volume for cell lines from several different types of human tissue. Brain (T98G, U-87), breast (MCF10A0JSB, HTERT-HME1, MDA-MB-231, MCF7, HCC1937a, T-47D), colorectal (SW480, SW620, HT-29, HCT116), lung (NCI-H2087, NL20, NCI-H2126), ovary (SK-OV-3, NIH: OVCAR-3, Coav3), pancreas (Panc-1, HTERT-HPNE, Capan-1), prostate (DU145, 22RV1, LnCaP Clone FG, PC-3, RWPE-1), skin (WM266–4, A375, MeWo, SK-MEL-2). (c) A model of eukaryotic cell growth with mass transport. During cell growth, amino acids are imported into the cytoplasm and assembled into non-ribosomal proteins and ribosomal proteins by mature ribosome particles. The ribosomal proteins are transported into the nucleoplasm to combine with ribosomal RNA to mature into ribosome particles, which are then transported back out to the cytoplasm. Non-ribosomal proteins are also actively transported in and out of the nucleoplasm. Proteins and ribosomes are also actively degraded. These processes can be captured by a simple set of mass flux equations.

proportion of the mature ribosome and ribosomal proteins is altered, such as during aneuploidy. The results show that the right proportion of mature ribosome particles is needed to maintain a high cell growth rate. By analyzing the gradient of growth rate and C/N ratio with respect to model parameters, it is found that to control cell growth rates, it is necessary to not only change amino-acid import and protein synthesis rates, but also modulate nucleoplasm-cytoplasm transport.

## A cell growth model with ribosome biogenesis and nucleoplasm-cytoplasm transport

Cell and nucleus volumes of a set of cell lines obtained for the PS-ON cell line collection are shown in Fig 1a and 1b (see Methods for experimental details). A schematic of the model is shown in Fig 1c. In this model, we consider essential processes in an eukaryotic cell, which includes protein synthesis, transport across various compartments, protein degradation and ribosome disassembly. For simplicity, we assume that all the molecules and particles are evenly distributed in cytoplasm and nucleus, so we can use a constant concentration to describe each component. In addition, we assume that the cell volume is proportional to the number of macro-molecules (including proteins and ribosomes), which is a good approximation for mammalian cells [14]. Under these assumptions, we can treat all the metabolic activities as chemical reactions in an isobaric and isothermal compartment. All the variables considered contain subscripts, which indicates the cytoplasmic ($C$) and nucleoplasm ($N$) compartments. 1) $P_{C,N}$ are the number of non-ribosomal proteins in the cytoplasm and nucleoplasm, respectively. 2) $A_C$, the number of amino acid molecules in the cytoplasm. $RP_{C,N}$ are the number of ribosomal proteins in the cytoplasm/nucleoplasm. $R_{C,N}$ are the assembled mature ribosome particles in the cytoplasm/nucleoplasm. The model variables and their order-of-magnitude estimates are given in Table 1.

### Model component 1: Protein synthesis

Protein synthesis is a sequential assembly process in the cytoplasm by mature ribosome particles. At each step, a new amino acid is added to a peptide chain (which is attached to the ribosome) and then the ribosome moves to the next codon. We assume that each step takes the same length of time and the total steps (average amino acid number) for non-ribosomal protein synthesis and ribosomal protein synthesis are $n_1$ and $n_2$, respectively. In a sequential assembly process, the synthesis rate is determined by the last step where $n_i - 1$ peptides become $n_i$ ($i = 1, 2$) peptides [15]. The synthesis rate is proportional to concentration of peptide chains with $n_i - 1$ ($i = 1, 2$) amino acids. If we assume there is sufficient tRNA, the synthesis rate should also be proportional to amino acid concentration. Accordingly, we can write the rate of

**Table 1. Glossary of model variables and their numerical estimates for new born cells.** See text for references.

| Symbol | Description | Yeast Values | Mammalian Values | Source |
|---|---|---|---|---|
| $A_C$ | Cytoplasmic amino acid (AA) number | $3200 \times 10^6$ | $21600 \times 10^6$ | [23, 37–39] |
| $P_C$ | Cytoplasmic non-ribosomal protein number | $15 \times 10^6$ | $660 \times 10^6$ | [33, 40, 45, 46] |
| $R_C$ | Cytoplasmic mature ribosome number | $0.135 \times 10^6$ | $6 \times 10^6$ | [33, 41] |
| $RP_C$ | Cytoplasmic ribosomal protein number | $1.35 \times 10^6$ | $60 \times 10^6$ | [33, 40, 45, 46] |
| $RP_N$ | Nuclear ribosomal protein number | $0.15 \times 10^6$ | $15 \times 10^6$ | [33, 40, 45, 46] |
| $R_N$ | Nuclear ribosome complex number | $0.015 \times 10^6$ | $1.5 \times 10^6$ | [33, 41] |
| $P_N$ | Nuclear non-ribosomal protein number | $1.65 \times 10^6$ | $165 \times 10^6$ | [33, 40, 45, 46] |
| $V_C$ | Cytoplasmic volume ($\mu m^3$ or fL) | 27 | 1500 | [34, 36, 43, 44] |
| $V_N$ | Nuclear volume ($\mu m^3$ or fL) | 3 | 500 | [34–36, 42–44] |

change in non-ribosomal protein number in terms of their synthesis rate in a container with volume $V_C$ as [16]:

$$\frac{dP_C}{dt} = s_1 [A_c][\Pi_{n-1}] V_C = s_1 \frac{A_C \Pi_{n_1 - 1}}{V_C} \tag{1}$$

where $P_C$ is cytoplasmic non-ribosomal protein number, $A_C$ is cytoplasmic amino acid number, $\Pi_{n_1 - 1}$ is the number of peptide chains in the last synthesis step, $V_C$ is the cytoplasm volume and $s_1$ is the synthesis rate coefficient (which depends on the protein mRNA, the ribosome translation speed and biochemical signaling networks that control synthesis). The brackets indicate their molar concentrations. $\Pi_{n_1 - 1}$ can be estimated as follows. First, the number of total peptide chains should be proportional to the number of ribosomes carrying out protein synthesis, which is proportional to the number of total mature ribosomes $R_C$. Second, we assume that peptide chain lengths in the ribosomes during active synthesis are uniformly distributed, the probability of finding peptide chains at the last step is $1/n_1$. Therefore on average, $\Pi_{n_1 - 1} \propto R_C/n_1$. The protein synthesis rate can be expressed as:

$$\frac{dP_C}{dt} = \frac{s_1 A_C R_C}{n_1 V_C} \tag{2}$$

Similarly, we can write the synthesis equation for ribosomal protein $RP_C$ (with synthesis rate coefficient $s_2$ and average amino acid number $n_2$ in a protein).

For ribosome assembly and maturation in the nucleus ($R_N$), we can assume there are several maturation sites (nucleolus), whose number is proportional to the nuclear volume ($V_N$), and the assembly should also be sequential. Therefore, similar to protein synthesis, we can write the ribosome assembly equation as:

$$\frac{dR_N}{dt} = \frac{s_3}{n_3} RP_N \tag{3}$$

where $RP_N$ is the ribosomal protein number in the nucleus, $n_3$ is the average ribosomal protein number in a ribosome and $s_3$ is the synthesis coefficient, including the number of synthesis sites and the assembly rate. According to the stoichiometric coefficients (Fig 1c), the rate of amino acid ($A_C$) and nuclear ribosomal protein ($RP_N$) loss due to synthesis can be written as:

$$\frac{dA_C}{dt} = - \frac{(s_1 + s_2) A_C R_C}{V_C} \tag{4}$$

$$\frac{dRP_N}{dt} = -s_3 RP_N \tag{5}$$

Note that the appearance of volume factors in Eqs (2) and (4) is because bimolecular reaction rates are proportional to reactant concentrations, or probabilities of reactants meeting in space (and also the overall compartment size), even when modeling changes in reactant numbers. Therefore, modeling of cell growth or cell mass accumulation necessarily involves the question of cell volume. Our model of cell and nuclear volumes is discussed in a later section.

## Model component 2: Transport

To synthesize proteins, the cell needs to import amino acids from the extracellular environment, and some of the imported amino acids go into the nucleus via free diffusion. After completing synthesis, cytoplasmic ribosomal proteins are transported into the nucleus to assemble

into ribosomal subunits. Then the two assembled ribosomal subunits are transported out of the nucleus, combine into a mature ribosome and participate in protein translation [7, 8]. For simplicity, in this model, we assume the complete ribosome is directly assembled in the cell nucleus. There is also bidirectional non-ribosomal protein transport across the nuclear envelope. Most of the macromolecule transport fluxes in the cell are active. For amino-acids in the cytoplasm, amino acid transporters (Slc gene family) on the cell surface are responsible for their import [17]. For protein transport across the nuclear envelope, import and export are carried out by the RanGTPase cycle [9].

In general, the transport rate depends on both the cargo concentration and transport protein density. For amino-acid import in our model, we will consider poor and rich nutrient conditions, corresponding to transporter saturated and cargo saturated conditions, respectively. For protein and ribosome transport across the nuclear envelope, we assume both the cargo and transporter proteins are unsaturated. If we assume that the transporter protein number ($TP_C$) is proportional to total protein number in cytoplasm $P_C$, we can obtain a general equation for active transport of protein and ribosome (when cargo concentration and transport protein density are both unsaturated):

$$j = k'[c][TP_C] = k[c]P_C/S \tag{6}$$

$$\frac{dc}{dt} = jS = k\frac{c}{V}P_C \tag{7}$$

Where $j$ is the cargo flux through membrane (per unit area per unit time), $[c]$, $[TP_C]$ and $[P_C]$ are concentration of cargo, transporter protein and cytoplasmic protein, respectively. $c$ is the total cargo number, $S$ is the surface area. The number of transport proteins is the concentration times the area of the nuclear envelope, which is proportional to $P_C$, i.e., $[TP_C]S = n_{TP_C} \propto P_C$. $k$ is the transport rate, which also includes the percentage of transport proteins. Based on this general equation, we can write all the transport equations as:

$$\frac{dA_C}{dt} = \frac{V_C}{V_C + V_N}\bar{t}_1, \tag{8}$$

$$\frac{dP_C}{dt} = -\left(\frac{t_4 P_C}{V_C} - \frac{t_5 P_N}{V_N}\right)P_C, \tag{9}$$

$$\frac{dR_C}{dt} = \frac{t_3 R_N P_c}{V_N}, \tag{10}$$

$$\frac{dRP_C}{dt} = -\frac{t_2 RP_C P_C}{V_C}, \tag{11}$$

$$\frac{dRP_N}{dt} = \frac{t_2 RP_C P_C}{V_C}, \tag{12}$$

$$\frac{dR_N}{dt} = -\frac{t_3 R_N P_C}{V_N} \tag{13}$$

$$\frac{dP_N}{dt} = \left(\frac{t_4 P_C}{V_C} - \frac{t_5 P_N}{V_N}\right)P_C \tag{14}$$

$P_N$ is the nuclear non-ribosomal protein number, $t_2$ is the import coefficient of cytoplasmic ribosomal protein, $t_3$ is the export coefficient of nuclear ribosome, $t_4$ and $t_5$ are import and export coefficients of non-ribosomal protein through the nucleus. $\bar{t}_1$ includes the exterior amino acid concentration and the import coefficient of amino acid. $\bar{t}_1 = $ constant when exterior amino acids are poor while $\bar{t}_1 = t_1 P_C$ when exterior amino acids are rich. For simplicity, we neglect the process of amino acid diffusion across nuclear envelop and assume that the imported amino acids are evenly distributed in cytoplasm and nucleoplasm depending on $V_C/V_N$.

In another model, Scott et al. [18] utilized a Michaelis-Menten form for the import coefficient of amino acids. Our high nutrient limit of the amino acid import coefficient gives the same results as the high amino acid concentration limit in Michaelis-Menten form. When the number of transport proteins is constant, The import rate coefficient saturates to a constant in high nutrient conditions, and becomes proportional to external amino acid concentration in low nutrient conditions.

## Model component 3: Degradation and disassembly

In this part, we consider protein degradation and ribosome disassembly, which occurs in both nucleus and cytoplasm [19]. The degraded proteins are broken down into short peptide chains and amino acids, and they are recycled in new protein translation. Similarly, ribosome disassembly produces new ribosomal proteins, which can also be reused in the ribosome synthesis cycle (Fig 1c). For simplicity, we assume that all the proteins only break down into amino acids, and these amino acids are evenly distributed in cytoplasm and nucleoplasm depending on the C/N ratio. There are two major pathways of protein degradation: the ubiquitin-proteasome pathway and the lysosomal proteolysis-mediate protein degradation [19, 20]. In both pathways, degradation is aided by degradation proteins or protein complexes. Therefore, the degradation rate is proportional to concentrations of degradation proteins and proteins to be degraded. Assuming the numbers of proteins assisting degradation in cytoplasm and nucleus are proportional to the numbers of non-ribosomal proteins in cytoplasm and nucleus respectively, we can get the general expression for protein degradation:

$$\frac{dn}{dt} = -k[P][DP]V = -k\frac{P}{V}DP \tag{15}$$

where $[P]$ and $[DP]$ are concentrations of proteins to be degraded and assisting degradation. Similarly, we can obtain the disassembly equation for ribosomes. Accordingly, we can write all the degradation and disassembly equations:

$$\frac{dP_C}{dt} = -d_{1C}\frac{P_C^2}{V_C}, \tag{16}$$

$$\frac{dR_C}{dt} = -d_{3C}\frac{R_C P_C}{V_C}, \tag{17}$$

$$\frac{dRP_C}{dt} = n_3 d_{3C}\frac{R_C P_C}{V_C} - d_{2C}\frac{RP_C P_C}{V_C}, \tag{18}$$

$$\frac{dRP_N}{dt} = -d_{2N}\frac{RP_N P_N}{V_N} + n_3 d_{3N}\frac{R_N P_N}{V_N}, \tag{19}$$

$$\frac{dR_N}{dt} = -d_{3N}\frac{R_N P_N}{V_N}, \tag{20}$$

$$\frac{dP_N}{dt} = -d_{1N}\frac{P_N^2}{V_N}, \tag{21}$$

$$\frac{dA_C}{dt} = \frac{V_C}{V_C + V_N} \left[n_1\left(\frac{d_{1C}P_C^2}{V_C} + \frac{d_{1N}P_N^2}{V_N}\right) + n_2\left(\frac{d_{2C}RP_CP_C}{V_C} + \frac{d_{2N}RP_NP_N}{V_N}\right)\right]. \tag{22}$$

$d_{1C}, d_{1N}$ are the degradation coefficients of non-ribosomal proteins in cytoplasm and nucleus. $d_{2C}, d_{2N}$ denote the degradation coefficients of ribosomal proteins. $d_{3C}, d_{3N}$ are the disassembly coefficients of ribosomes. These coefficients include both the degradation rate and percentage of degradation proteins assisting the process.

Putting all three parts together, we obtain the governing equations for the dynamics of the complete proteome (See parameter details in Table 2):

$$\frac{dA_C}{dt} = \frac{V_C}{V_C + V_N} \left[\bar{t}_1 + n_1\left(\frac{d_{1C}P_C^2}{V_C} + \frac{d_{1N}P_N^2}{V_N}\right) + n_2\left(\frac{d_{2C}RP_CP_C}{V_C} + \frac{d_{2N}RP_NP_N}{V_N}\right)\right] - \frac{(s_1 + s_2)A_CR_C}{V_C}, \tag{23}$$

$$\frac{dP_C}{dt} = \frac{s_1A_CR_C}{n_1V_C} - \frac{d_{1C}P_c^2}{V_C} - \left(\frac{t_4P_C}{V_C} - \frac{t_5P_N}{V_N}\right)P_C \tag{24}$$

$$\frac{dR_C}{dt} = \frac{t_3R_NP_C}{V_N} - \frac{d_{3C}R_CP_C}{V_C} \tag{25}$$

**Table 2. Model parameters and their numerical estimates.**

| Parameter | Description | Yeast | Mammalian | Source |
|---|---|---|---|---|
| $r_{1,2}$ | Macromolecule number and cytoplasmic (nuclear) volume ratio ($10^6/\mu m^3$) | 0.6 | 0.6 | Estimate |
| $n_1$ | Average number of AA in non-ribosomal proteins | 400 | 400 | [48] |
| $n_2$ | Average number of AA in ribosomal proteins | 400 | 400 | [48] |
| $n_3$ | Average number of protein subunits in mature ribosomes | 80 | 80 | [49] |
| $t_1$ | Amino acid import rate coefficient ($h^{-1}$) | 400 | 36 | Estimate |
| $t_2$ | Ribosomal protein transport coefficient across nuclear envelop ($\mu m^3/10^6 h$) | 9 | 1 | [50] |
| $t_3$ | Ribosome transport coefficient across nuclear envelop ($\mu m^3/10^6 h$) | 1.2 | 0.15 | [50] |
| $t_4$ | Non-ribosomal protein transport coefficient into nucleus ($\mu m^3/10^6 h$) | 0.8 | 0.1 | Estimate |
| $t_5$ | Non-ribosomal protein transport coefficient out of nucleus ($\mu m^3/10^6 h$) | 0.75 | 0.05 | Estimate |
| $d_{1C}, d_{1N}$ | Non-ribosomal protein degradation coefficient ($\mu m^3/10^6 h$) | 0.1 | 0.01 | Estimate |
| $d_{2C}, d_{2N}$ | Ribosomal protein degradation coefficient ($\mu m^3/10^6 h$) | 0.1 | 0.01 | Estimate |
| $d_{3C}, d_{3N}$ | Ribosome disassembly coefficient ($\mu m^3/10^6 h$) | 0.1 | 0.01 | Estimate |
| $s_1$ | Non-ribosomal protein synthesis coefficient ($\mu m^3/10^6 h$) | 155 | 115 | [33, 45, 46] |
| $s_2$ | Ribosomal protein synthesis coefficient ($\mu m^3/10^6 h$) | 125 | 95 | [33, 45, 46] |
| $s_3$ | Ribosome assembly coefficient ($h^{-1}$) | 48 | 1.5 | [33] |

$$\frac{dRP_C}{dt} = \frac{n_3 d_{3C} R_C P_C}{V_C} + \frac{s_2 A_C R_C}{n_2 V_C} - \frac{t_2 RP_C P_C}{V_C} - \frac{d_{2C} RP_C P_C}{V_C} \tag{26}$$

$$\frac{dRP_N}{dt} = \frac{t_2 RP_C P_C}{V_C} - s_3 RP_N - \frac{d_{2N} RP_N P_N}{V_N} + \frac{n_3 d_{3N} R_N P_N}{V_N} \tag{27}$$

$$\frac{dR_N}{dt} = \frac{s_3}{n_3} RP_N - \frac{t_3 R_N P_C}{V_N} - \frac{d_{3N} R_N P_N}{V_N} \tag{28}$$

$$\frac{dP_N}{dt} = \left( \frac{t_4 P_C}{V_C} - \frac{t_5 P_N}{V_N} \right) P_C - \frac{d_{1N} P_N^2}{V_N} \tag{29}$$

where $A_C$, $P_{C,N}$, $R_{C,N}$, $RP_{C,N}$ are cytoplasmic/neoplasmic number of amino-acids, non-ribosomal proteins, ribosomes and ribosomal proteins, respectively. $n_1$, $n_2$ are average amino acid numbers per non-ribosomal and ribosomal protein. $n_3$ is the average number of proteins in a ribosome. $\bar{t}_1$ is the amino acid uptake rate, which includes the exterior amino acid abundance, transport efficiency and percentage of transport proteins. $\bar{t}_1 = $ const when exterior amino acid is insufficient while $\bar{t}_1 = t_1 P_C$ when amino acid is sufficient. $t_2$, $t_3$, $t_4$, $t_5$ are transport coefficients of $RP_C$, $R_N$, $P_C$, $P_N$, which include transport efficiency and percentage of transport protein. $d_{1C}$, $d_{2C}$, $d_{1N}$, $d_{2N}$, $d_{3C}$, $d_{3N}$ are degradation or disassembly coefficients, which include degradation efficiency and percentage of degradation (assisting) proteins. $s_1$ and $s_2$ are synthesis coefficients of non-ribosomal and ribosomal protein, which include mRNA concentration and ribosome moving speed. $s_3$ is the assembly coefficient of ribosome, which includes the number of assembly sites and assembly rate at each site. $V_N$ and $V_C$ are nucleus and cytoplasm volumes, respectively.

## On the mapping between molecular content and cell and nucleus size

Eqs 23–29 are the general equations for the time rate of change of protein, ribosomal protein, ribosome particles and amino acids in connected cytoplasm and nucleoplasm compartments with volumes $V_C$ and $V_N$, respectively. The remaining important question is how are $V_C$ and $V_N$ determined. The largest number of molecules in the cell is water, followed by ions [21]. The number of amino acids and proteins are smaller in comparison, although in yeast the amino-acid concentration is comparable to ionic concentration [22, 23]. From studies on cell ion regulation, it is found that by actively passaging ions across the cell surface, cells maintain a constant osmolarity difference between the cytoplasm and the extracellular environment [24–26]. This ion homeostasis model may be generalized to include charged small organic molecules such as some amino-acids, whose import must rely on transporters that are either sensitive to membrane voltage, or co-transporters that import these charged molecules with their associated counter ions (Fig 2) [17, 27]. Therefore, it is reasonable to consider ions, amino-acids and other charged small molecules as part of the cell ionic homeostasis system.

As a consequence of the cell ion homeostasis, the water content is proportional to the total number of *impermeable* molecules such as proteins and buffers in the cytoplasm [26, 28]. Ions and small amino acids are thought to freely diffuse across the nuclear envelope according to their concentration gradients. Therefore, at equilibrium, there is no ion and small molecule concentration differences between the nucleoplasm and cytoplasm, but there could be a protein concentration difference.

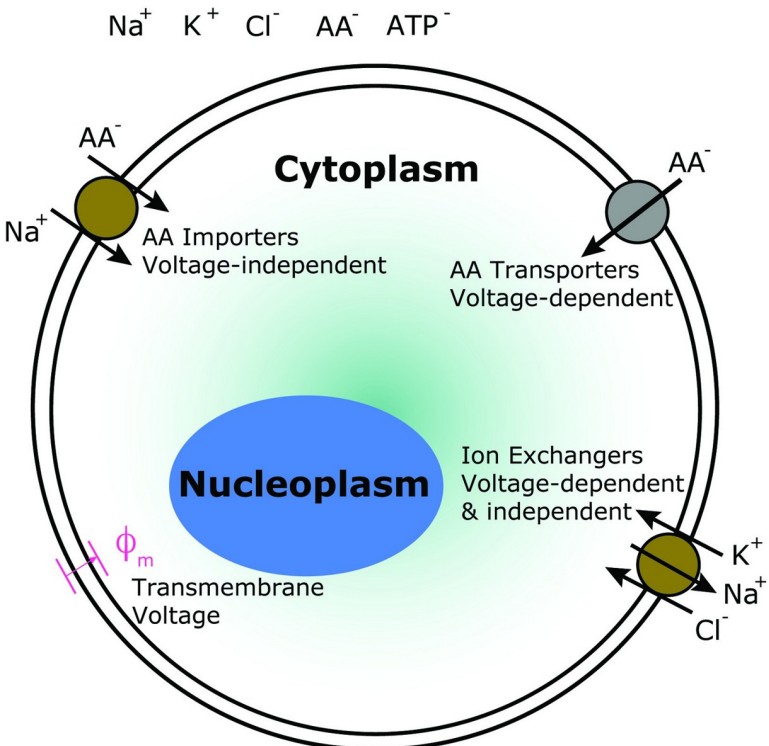

**Fig 2. The cell volume is proportional to the impermeable macromolecular number (proteins, buffers, and ribosomes) because the concentration of small charged molecules such as amino acids and ATP are likely regulated together with concentration of ions.** For example, amino acids can be imported into the cytoplasm together with Na$^+$ in a voltage independent manner, or imported individually in a voltage dependent manner. The transmembrane voltage $\phi_m$ also depends on ion concentration (Na$^+$, K$^+$, Cl$^-$, etc). Models of cell ionic homeostasis predict that the cell water content is directly proportional to cell macromolecular number.

If we accept that the concentrations of small molecules and ions are regulated together, the it is reasonable to assume that the cell volume is proportional to the protein and macromolecular number.

$$V_C = (P_C + R_C + RP_C)/r_1 \tag{30}$$

where $r_1$ is a proportionality constant that may depend on the ionic regulation system and the overall osmotic pressure of the cytoplasm. The total osmotic pressure difference between the cytoplasm and the external environment also should depend on the cytoplasmic hydraulic pressure, because at equilibrium the osmotic pressure difference is equal to the hydraulic pressure difference. The hydraulic pressure, in turn, may depend on cytoskeletal contractility, and therefore $r_1$ may depend on other variables such as cytoskeletal activity.

For the nucleoplasm, we have

$$V_N = (P_N + R_N + RP_N)/r_2 \tag{31}$$

Here we see that because the total osmolarity of the nucleus may be different than the cytoplasm, there might be an osmotic gradient and hydraulic pressure gradient across the nuclear envelope. This excess pressure in the nucleus must be mechanically balanced by the tension in the nuclear envelope and/or cytoskeleton activity. Indeed, for mammalian cells, F-actin and associated contractile forces are linked to the nuclear lamins through the LINC complex [29]. Therefore, there

could be mechanical tension in the nuclear envelope. Moreover, intermediate filaments such as vimentin also surrounds the nucleus and may provide additional mechanical support [30]. Therefore, the proportionality constants, $r_{1,2}$, between the volume of the compartment and the macromolecular content could be different. The nuclear compartment also contains DNA, which in principle also contributes to volume via contributing to nuclear osmotic pressure. However, the overall DNA mass ($\sim 6$ pg [31]) is small when compared to proteins and ribosomes ($\sim 150$pg in total and $\sim 30$pg in nucleus [32]). DNA also has a larger molecular weight ($\sim 650$kDa for a 1000 bp DNA strand) and a smaller concentration than proteins (40kDa). Consequently, we expect DNA to contribute less than proteins to the overall osmolarity and the nuclear volume. Therefore, we neglect the DNA contribution to the nuclear volume here.

Another complicating feature of the volume model is that the nuclear ribosomes are not well mixed with the nucleoplasm, and instead are phase separated into nucleolus. The contribution of nucleolus to the nuclear volume is unclear, and a different model utilizing the volume of phase separated material might be needed. Such a model may be different in numerical detail, but the overall nuclear volume should still be generally proportional to the volume of phase separated nucleolus.

## Parameter estimation

All initial parameter estimates are based on *Saccharomyces cerevisiae*. The cell cycle time for yeast is around 2 hours [33]. A typical cell volume is 40 $\mu$m$^3$ [34], and cell to nuclear volume ratio (C/N ratio) is around 10 [35]. Assuming that mature cells are 60 $\mu$m$^3$ [36] and newborn cells grow exponentially from half of the mature size: 30 $\mu$m$^3$, we can obtain a rough estimation for cell volume during a cell cycle: $V_C \approx 30e^{0.35t}$ $\mu$m$^3$. We can also obtain an average cell size: $\bar{V} = \int_0^{t_d} V(\tau)d\tau/t_d \approx 1.3V_b$, where $V_b$ is the volume of a newborn cell. The following calculations will be based on this average cell state (All components are assumed proportional to cell volume).

For mammalian cells, fewer parameters are available from literature. Here we scale up the nucleoplasm-cytoplasm transport parameters ($t_2$, $t_3$) from yeast. Other parameters are based on literature. The rescaled parameters values are close to literature data (See details in the S1 Text).

## Initial conditions

Intracellular concentration of amino acids could vary from 100 to 500mM for yeast cells [23, 37]. In our estimate, we set the concentration to be 200mM, so the total number of amino acids for a newborn cell is: $A_C \approx 3.2 \times 10^9$. The ribosome number for a newborn cell is around $0.15 \times 10^6$ [33]. Assuming the ribosome numbers in cytoplasm and nucleus are proportional to cytoplasm and nucleus volume, we obtain: $R_C = 0.135 \times 10^6$, $R_N = 0.015 \times 10^6$. A mature yeast cell has a total of around $6 \times 10^7$ proteins [45], which contains both ribosomal and non-ribosomal proteins. Therefore, for a new born cell, we have approximately 1/2 of the mature cell: $n_3R_C + RP_C + P_C = 27 \times 10^6$ for cytoplasmic proteins, and $n_3R_N + RP_N + P_N = 3 \times 10^6$ for nuclear proteins, where $n_3 = 80$ is the average number of proteins in a mature ribosome particle. Among all the proteins, ribosomal protein percentage could vary from 30% to 50% [33, 46]. In our estimate, we set the fraction to be 45%. Assuming the amount in the cytoplasm is 9 times of that in the nucleus, we have: $RP_C = 0.45 \times 27 \times 10^6 - 80 \times 0.135 \times 10^6 \approx 1.35 \times 10^6$, $P_C = 27 \times 10^6 - 80 \times 0.135 \times 10^6 - 1.35 \times 10^6 = 1.5 \times 10^7$, $RP_N = 0.15 \times 10^6$, $P_N = 1.65 \times 10^6$. The yeast parameters also agree with literature [47].

The conversion factor between macromolecule number and volume therefore, is: $(R_C + RP_C + P_C)/V_C \approx 0.6 \times 10^6/\mu$m$^3$. We assume for both yeast and mammalian cells, this

conversion factor is the same for the nuclear compartment. The initial conditions for both yeast and mammalian cells are listed in Table 1.

## Kinetic parameters

We first consider parameters in synthesis. We assume that the average numbers of amino acids in each ribosomal protein and non-ribosomal protein are the same ($n_1 = n_2$), which is around 400 [48]. Roughly speaking, a yeast cell must produce 2000 ribosomes per min [33] to maintain metabolism, which means: $s_3/n_3 RP_N = s_3/80 * 0.15 \times 10^6/0.75 \approx 0.12 \times 10^6 \text{h}^{-1}$, so $s_3$ = 48 h$^{-1}$. There is a factor of 0.75 because we are considering the average cell state. Assuming newborn cells have half of the mature cell proteins, the cell needs to synthesize $3 \times 10^7$ proteins in a single cell cycle, thus: $((s_1 + s_2)A_C R_C)/(n_1 V_C) = 1.5 \times 10^7 \text{h}^{-1}$, so $s_1 + s_2 = 280 \mu\text{m}^3/10^6\text{h}$. Synthesis rate constant should be positively correlated to the amount of each component, so for simplicity, we assume $s_1 : s_2 = P : RP = 11 : 9$. Then we obtain: $s_1 = 155$, $s_2 = 125 \mu\text{m}^3/10^6\text{h}$.

For parameters in transport, we first consider the amino acid uptake rate in rich-nutrient (sufficient amino acids) case. The imported amino acids are utilized in two parts: synthesizing proteins and maintaining the amino acid pool concentration, so $t_1 P_C = n_1((s_1 + s_2)A_C R_C)/(n_1 V_C) + (A_C(t) - A_C(0))/2\text{h} = 6000 + 1800$. Therefore: $t_1 \approx 400 \text{h}^{-1}$. During the ribosome assembly process, a rapidly growing yeast cell must import $\sim 150000$ ribosomal proteins per minute across the nuclear envelope, and export $\sim 4000$ ribosomal subunits (equivalently 2000 mature ribosomes) per minute [50], which means: $t_2 RP_C P_C/V_C = 9 \times 10^6/\text{h}$, $(t_3 R_N P_C)/V_N = 0.12 \times 10^6/\text{h}$, so $t_2 = 9 \mu\text{m}^3/10^6\text{h}$, $t_3 = 1.2 \mu\text{m}^3/10^6\text{h}$. For non-ribosomal protein transport through the nuclear envelope, we assume that the transport rate is equal to that of ribosomal protein: $t_2 RP_C P_C/V_C = (t_4 P_C^2)/V_C$, so $t_4 = 0.8 \mu\text{m}^3/10^6\text{h}$. The net import should be positive, so we obtain $t_5 = 0.72$, which is slightly lower than $t_4$.

For protein degradation and ribosome disassembly, we assume ribosome disassembly and protein degradation rates in rapidly growing cells are 1/10 of the synthesis rate: $(d_{1C}P_C^2/V_C + d_{2C}RP_C P_C/V_C + d_{1N}P_N^2/V_N + d_{2N}RP_N P_N/V_N)/((s_1 + s_2)A_C R_C/(n_1 V_C)) = 1/10$, $(d_3 CR_C P_C/V_C + d_3 NR_N P_N/V_N)/(s_3/n_3 RP_N) = 1/10$. Assuming all the degradation rates are equal, we can get: $d_{1C} = d_{2C} = d_{1N} = d_{2N} = 0.1 \mu\text{m}^3/10^6\text{h}$, $d_{3C} = d_{3N} = 0.1 \mu\text{m}^3/10^6\text{h}$. This result is in the reasonable range given in literature [51] (converted into $d \approx 0.01 - 2(\mu\text{m}^3/(10^6\text{h}))$ in our model parameters). A summary of all estimated kinetic parameters is shown in Table 2.

For mammalian cells, the cell volume is roughly 100 times larger, and the C/N ratio is around 3–10. Typical cell cycle time is around 20 hours. Therefore, the parameters are different. Our estimates for the parameters for a prototypical mammalian cell are also given in Table 2.

## Results

### Exponential vs linear growth are determined by amino acid import

Using the proposed models of $V_C$ and $V_N$ in Eqs (30) and (31) and the estimated parameters, we can compute the change in cell protein content, ribosome content, nuclear volume, and the cell volume during a complete cell cycle for both yeast and mammalian cells in different conditions (e.g. rich amino acid, poor amino acid, and quiescent cases). In the simulation, the cell divides when the cell volume reaches twice the initial cell volume $V_0$. After division, all the components are set to 1/2 of the mother cell number. We note that for *S. cerevisiae*, the mother cell grows a bud, and the volume of the bud after pinching off from the mother cell can be variable. At optimal nutrient conditions, the daughter cell volume is approximately 1/2 of the mother cell [52]. The division mechanism in simulation does not effect the cell growth rate or

the computed cell to nucleus volume ratio. In the case where the amino acid import rate is proportional to the protein number, i.e., $\bar{t}_1 \propto P_C$, our model predicts exponential growth of cell components in terms of their overall number, and exponential growth of cell and nuclear volume (Fig 3a, 3b, 3e and 3f)), i.e., all quantities are proportional to $e^{\lambda t}$, where $\lambda$ is the cell growth rate. For these parameters, the majority of the cell consists of non-ribosomal proteins, although ribosomal proteins is a substantial fraction of the overall proteome.

Fig 3c and 3g show cell growth trajectories in rich amino acid, poor amino acid and non-growing (quiescent) conditions. In the nutrient-limited case, the transport rate is set to be a constant, $\bar{t}_1 = 2850 \, \mathrm{h}^{-1}$ for the yeast cell and $\bar{t}_1 = 15000 \, \mathrm{h}^{-1}$ for the mammalian cell. In the stationary case, the cell is usually in an amino-acid-limited environment. Moreover, protein synthesis is decreased and protein degradation is increased, which is a common feature for both yeast [53] and mammalian cells [54–56]. In this case, we assume that proteasomes and lysosomes are sufficient and the degradation (disassembly) rate is only proportional to the number of proteins or ribosomes to be degraded (disassembled) (i.e., $dP/dt = -d \times P$, $dR/dt = d \times R$). All the degradation coefficients are set to be $d = 0.8$ for yeast cell and $d = 0.05$ for mammalian cell. For yeast cell, the transport rate is $\bar{t}_1 = 2850 \, \mathrm{h}^{-1}$ and ribosome synthesis rate is set to be $s_3 = 5$ ($\bar{t}_1 = 15000 \, \mathrm{h}^{-1}$ and $s_3 = 0.3$ for mammalian cell). All the other parameters (e.g., cytoplasm-nucleoplasm transport coefficients) are the same in all three cases above. Compared with the cell in rich-nutrient environment (exponential growth), in poor-nutrient environment, our model predicts that the cell grows almost linearly due to limited supply of amino acids. This linear growth is a good approximation for steady growth after around 10 cell cycles (S1 Text) In the stationary case, the cell almost maintains a constant volume due to the balance between synthesis and degradation.

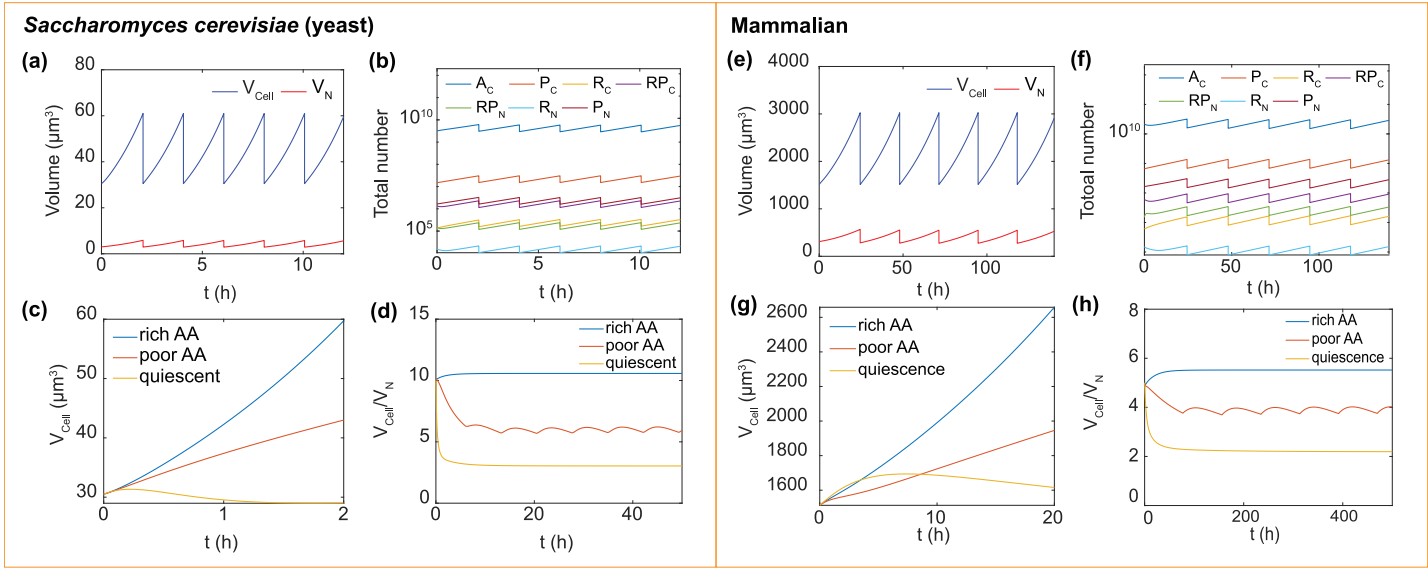

**Fig 3. Model results for cell and nuclear volumes of *Saccharomyces cerevisiae* (yeast) and HeLa (mammalian) over several cell cycles.** Here the cell divides when the cell volume doubles. (a) The cell volume ($V_{Cell} = V_C + V_N$) and nuclear volume ($V_N$) for a repeatedly dividing cell. (b) The number of cell components (amino acids, proteins, ribosomal proteins and ribosomes) for successive rounds of cell division. (c) The cell volume grows exponentially if amino acid is rich, and $\bar{t}_1 \propto P_C$. The computed growth trajectory is approximately linear if amino acid is poor and $\bar{t}_1$ is a constant. In the quiescent case, the cell volume becomes stationary. (d) In all cases, the C/N ratio ($V_{Cell}/V_N$) reaches a constant. In the quiescent case, the cell growth rate is zero, and the C/N ratio is small. (e)-(h) Corresponding results for mammalian cell. Parameters used are listed in Tables 1 and 2.

## Cell to nucleus volume ratio

Consistent with observations, our model predicts a constant cell to nucleus volume (C/N) ratio for all cases, but the value of the ratio varies significantly for different cells in different conditions. Fig 3d and 3g show the change of cell to nucleus volume ratio (C/N ratio) in several cell cycles in the three cases. For both rich-amino acid and quiescent cases, C/N ratios reach a constant (10 and 3 for yeast cell; 5.5 and 2.2 for mammalian cell). The C/N ratio is low in quiescent cell because there are relatively more proteins in nucleus due to nuclear import and decreased ribosome synthesis. For the nutrient-limited cell, the C/N ratio fluctuates periodically. This is because in one cell cycle, the linearly growing cell is unable to adjust all the components into a stable state. At the beginning of the cycle, C/N ratio increases, then it decreases because imported amino acids are not sufficient to maintain increased cytoplasmic protein synthesis. This result comes from the assumption that amino acid import is constant throughout the cell cycle, but it is also possible that the cell adjusts amino acid import as the cell cycle progresses. After each division, the cell state is reset. Notably, in the three cases above, protein distributions in cytoplasm and nucleoplasm are totally different due to different synthesis and transport, and this is a major reason for the different C/N ratios in these cases.

## Effects of protein transport and synthesis on cell growth rate

Fig 4a–4e and 4k–4o show how model parameters influence the cell growth rate, λ, in exponentially growing cells (rich-amino acid case) for both yeast and mammalian cells. The amino acid import coefficient $t_1$ increases the growth rate (Fig 4a and 4k). When $t_1$ is low, the growth rate can be negative because there is more protein degradation than synthesis. The result shows that for both types of cells, the growth rate does not plateau with increase of $t_1$. In reality, however, there should be an upper limit for amino acid transport rate ($t_1$). Ribosomal protein and ribosome transport coefficients $t_2$ and $t_3$ both increase the growth rate when they are small. However, when they are large, $t_3$ has little influence, while $t_2$ decreases the growth rate (Fig 4b and 4l)). This is because when $t_2$ is large, the number of cytoplasmic ribosomal protein is smaller, which decreases the cytoplasmic volume. In the meantime, although more ribosomal proteins are transported into the nucleus, the ribosome assembly process decreases the total number of molecules in nucleus (80 ribosomal proteins turn into 1 ribosome), and the nuclear volume does not increase. Overall, larger $t_2$ causes a decrease in cell volume, and thus the growth rate. Compared with other transport activities, transport of non-ribosomal proteins into ($t_4$) and out of ($t_5$) nucleus is more important. There is a sharp transition around the line $t_4 = t_5$. when $t_5$ is greater than $t_4$, the growth rate keeps constant; otherwise, the growth rate declines significantly (Fig 4c and 4m). This sharp change is because: Increase of $t_4$ significantly decreases the cytoplasmic protein amount, thus decreasing the transport-related proteins in the cytoplasm proportionally. As a result, the ribosome synthesis cycle is slowed down, thus limiting the growth rate. On the other hand, with increasing $t_5$, the transported proteins accumulate. However, the growth rate is already "saturated", so accelerating the ribosome cycle influences the growth rate very little. This result implies that the cell has to accurately control protein transport across the nuclear envelope.

Non-ribosomal protein synthesis coefficient $s_1$ and ribosome assembly coefficient $s_3$ both increase the growth rate. However, $s_1$ has a more obvious influence than $s_3$ (Fig 4d and 4n). The ribosomal protein synthesis coefficient $s_2$ also plays an important role in cell growth. Both synthesis coefficients $s_1$ and $s_2$ increase the growth rate. However, there is also a trade-off between $s_1$ and $s_2$, which is indicated by the non-monotonicity of growth rate when fixing $s_1 + s_2$ (Fig 4e and 4o). Therefore, there is an optimal ribosomal and non-ribosomal protein

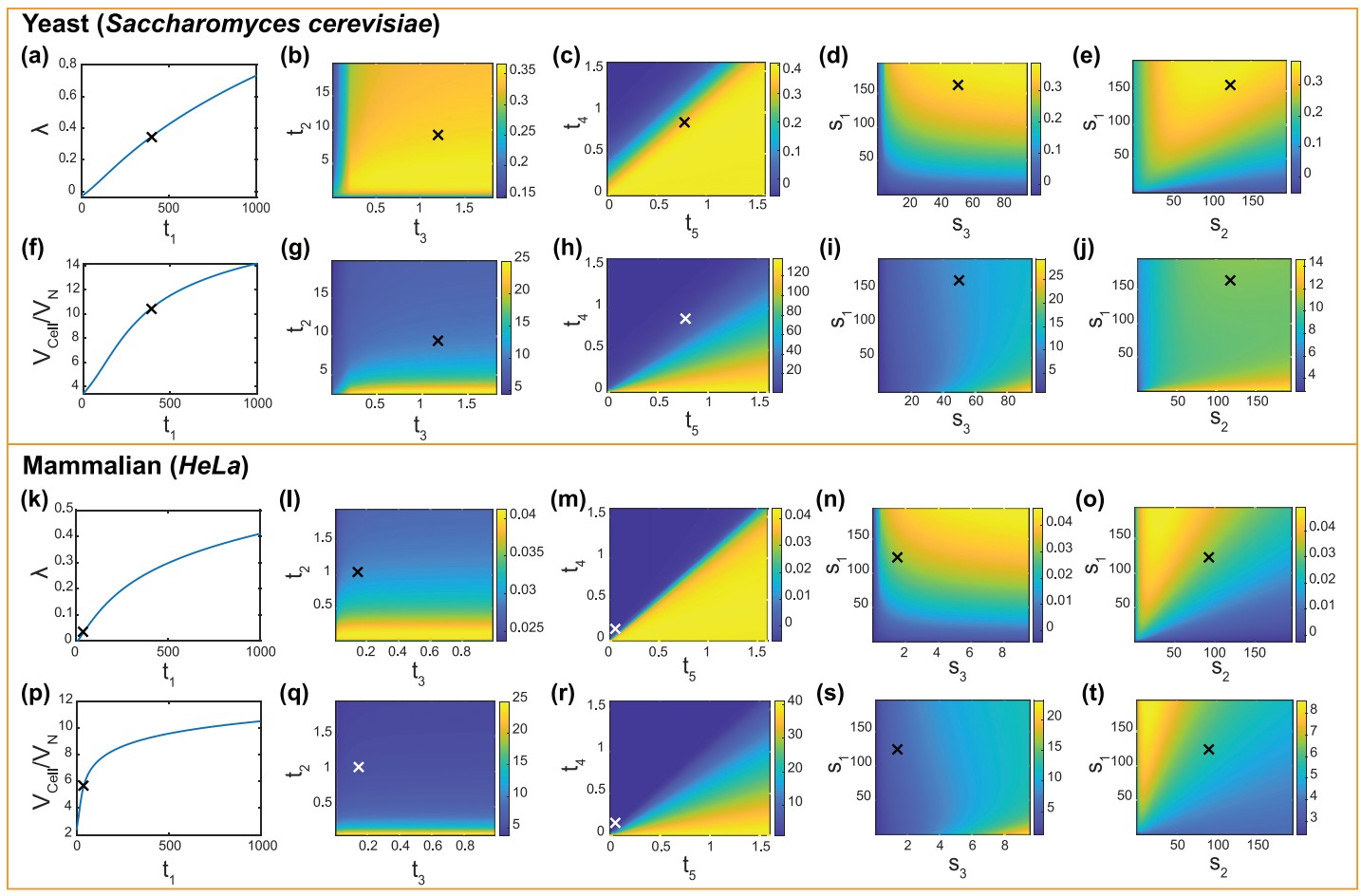

**Fig 4. Effects of synthesis and transport parameters on the cell growth rate and the C/N ratio.** (a)-(c) Effects of amino acid import ($t_1$), ribosomal protein transport ($t_2$), ribosome transport ($t_3$) and non-ribosomal protein transport ($t_4$ and $t_5$) on growth rate. Notably, the growth rate is non-monotonically influenced by $t_2$, and there's a sharp change in growth rate around $t_4 = t_5$. (d)-(e) Effects of non-ribosomal and ribosomal protein synthesis ($s_1$ and $s_2$) and ribosome synthesis ($s_3$) on growth rate. There is an optimal $s_1/s_2$ for the maximum growth rate. (f)-(j) Effects of transport and synthesis on the C/N ratio. protein synthesis coefficients $s_1$ and $s_2$ have a non-monotonic influence on C/N ratio. (k)-(t) Corresponding results for mammalian cells. Interestingly, compared to yeast cell case, $s_1$ and $s_2$ have different effects on C/N ratio due to different ribosome synthesis coefficient $s_3$. Realistic values are marked as "X". All parameters except for the scanned parameters are listed in Table 2.

synthesis ratio for the maximum growth rate. This non-monotonicity also agrees with findings in bacteria [18].

## Effects of protein transport and synthesis on C/N ratio

Fig 4f–4j and 4p–4t show how model parameters influence the C/N ratio for both yeast and mammalian cells. In exponentially growing cells, amino acid uptake coefficient $t_1$ increases the C/N ratio because the amino acid import rate is much more than the protein transport rate across nuclear membrane, thus causing the accumulation in the cytoplasm (Fig 4f and 4p). The ribosome transport coefficient $t_3$ increases the C/N ratio when it is small, but has little influence when large (Fig 4g and 4q). In contrast, the ribosomal protein transport coefficient $t_2$ decreases the C/N ratio and it is more significant (Fig 4g and 4q). Similar to growth rate, the non-ribosomal protein transport coefficients across nuclear membrane (into: $t_4$, out of: $t_5$) also influence the C/N ratio significantly, especially around $t_4 = t_5$ (Fig 4h and 4r).

Interestingly, in addition to transport, synthesis also influences the C/N ratio. When the ribosome assembly rate coefficient $s_3$ is high, the ribosomal proteins are rapidly converted to ribosomes, causing a rapid decrease in macromolecule number in the nucleus, thus resulting in a small nuclear volume and high C/N ratio (Fig 4i and 4s). For yeast and mammalian cells, non-ribosomal protein synthesis coefficient $s_1$ and ribosomal protein synthesis coefficient $s_2$ have different effects on the C/N ratio due to different $s_3$ (Fig 4j and 4t). For yeast cells, $s_3$ is large, and ribosomal proteins in the nucleus are assembled at a high rate. Therefore, nuclear ribosomal proteins only take up a small portion of the nuclear volume. When increasing $s_2$, $RP_C$ increases much more than $RP_N$, so $s_2$ increases the C/N ratio. For a realistic ribosomal synthesis rate ($s_2 = 125$), $s_1$ decreases C/N ratio. When $s_1$ is small, ribosomal proteins dominate the cell volume. Due to large $s_3$, the C/N ratio is high. However, when $s_1$ is large, non-ribosomal proteins take a major part, and the influence of $s_3$ on the C/N ratio is small. Non-ribosomal proteins are rapidly imported into the nucleus and cause a decrease in the C/N ratio. However, for mammalian cells, $s_3$ is much smaller than for yeast, therefore $s_1$ and $s_2$ have opposite effects on the C/N ratio.

## Effects of degradation and disassembly

We first consider the influence of protein degradation on the C/N ratio and the cell growth rate. Non-ribosomal protein degradation ($d_1$) monotonically decreases growth rate and increases C/N ratio for both yeast and mammalian cells (Fig 5a and 5e). Ribosomal protein degradation ($d_2$) has little influence on the growth rate. However, $d_2$ decreases the C/N ratio for mammalian cell when $d_1$ is large (Fig 5b and 5f).

In contrast from protein degradation, ribosome disassembly has two effects on cell growth: on the one hand, the disassembly of ribosome produces more ribosomal proteins ($R_C \rightarrow n_3 RP_C$), thus increasing the cell volume; on the other hand, disassembly decreases the protein synthesis rate. With competition between these two effects, we see that as the ribosomal and non-ribosomal protein ratio varies, disassembly has different influences on cell growth (For simplicity, we assume the degradation rates are the same in cytoplasm and nucleus. i.e., $d_{3C} = d_{3N} = d_3$). For both yeast and mammalian cells, for high non-ribosomal

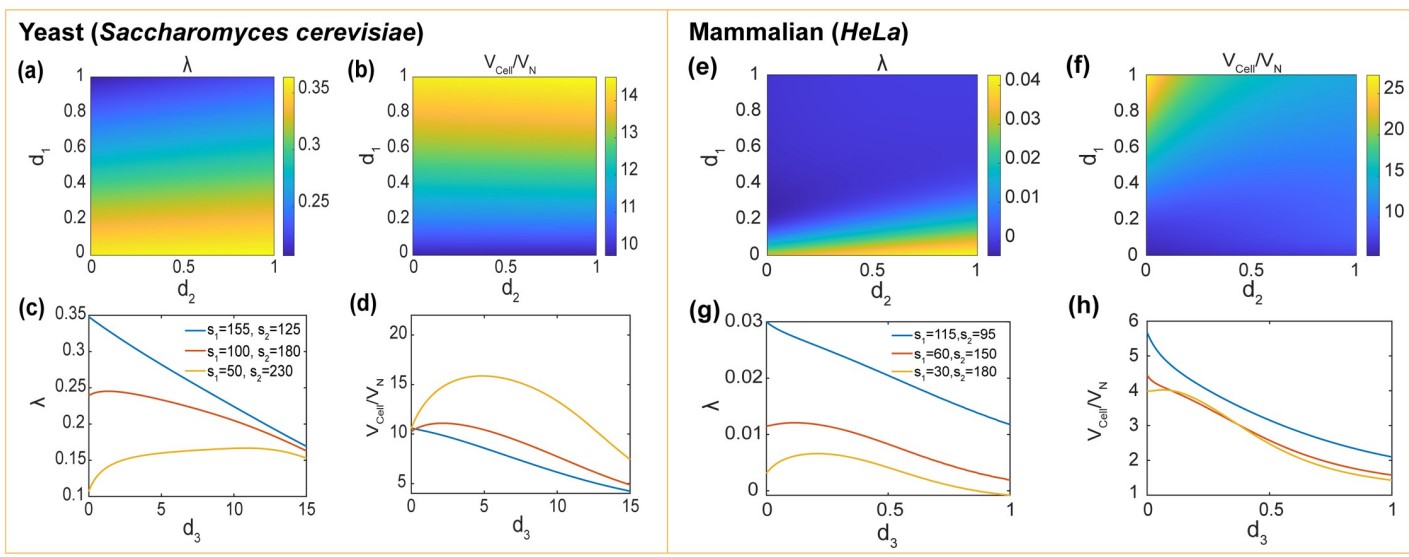

**Fig 5. Effects of protein degradation and ribosome disassembly on cell growth and the C/N ratio.** (a)-(b) effects of degradation on growth rate and C/N ratio. (c)-(d) Non-monotonic influences of disassembly on growth rate in different ribosomal protein synthesis levels. (e)-(h) Corresponding results for mammalian cells. All parameters except scanned parameters are listed in Table 2.

protein ratio: $s_1 = 155$, $s_2 = 125$ for yeast cell and $s_1 = 115$, $s_2 = 95$ for mammalian cell, $d_3$ decreases the growth rate monotonically. However, when $s_2/s_1$ is increased, $d_3$ changes the growth rate non-monotonically (Fig 5c and 5g). When $d_3$ is low, it increases the growth rate; when it is high, it decreases the growth rate. These results can be explained as follows: When $s_1/s_2$ is high, non-ribosomal protein dominates the cell volume. With increase of ribosome disassembly, ribosome number decreases, thus decreasing the non-ribosomal protein synthesis rate, and this influence is larger than the volume increase caused by $RP$ increase. However, when $s_1/s_2$ is low, the ribosomal protein amount increases and also becomes important for determining cell volume, and the increase of $RP$ number influences more than the growth decrease caused by less ribosomes. It is worth noting that when $d_3$ is too high, it also decreases the ribosomal protein amount due to insufficient ribosomes. As a result, $d_3$ finally decreases the growth rate. Ribosome disassembly has similar influence on the C/N ratio (Fig 5d and 5h).

### Effects of ribosomal protein to ribosome ratio

It is suggested that in aneuploid cells, cell volume abnormality might be related to the abnormal ratio of protein complex amount to free protein amount [57]. The ribosome is a major protein complex in the cell, and the ratio of ribosome to ribosomal protein might be an indicator of aneuploidy. Therefore, it is necessary to explore how the ratio RP/R influences cell growth. In our model, there are several ways to tune this ratio: Generally, $d_3$ increases the ratio, while $s_3$ or $t_2$ decreases this ratio. When tuning $d_3$, the effect of RP/R on growth rate is similar to that of $d_3$ (Fig 6a and 6d). In the process of tuning $s_3$, when RP/R is low ($s_3$ is high),

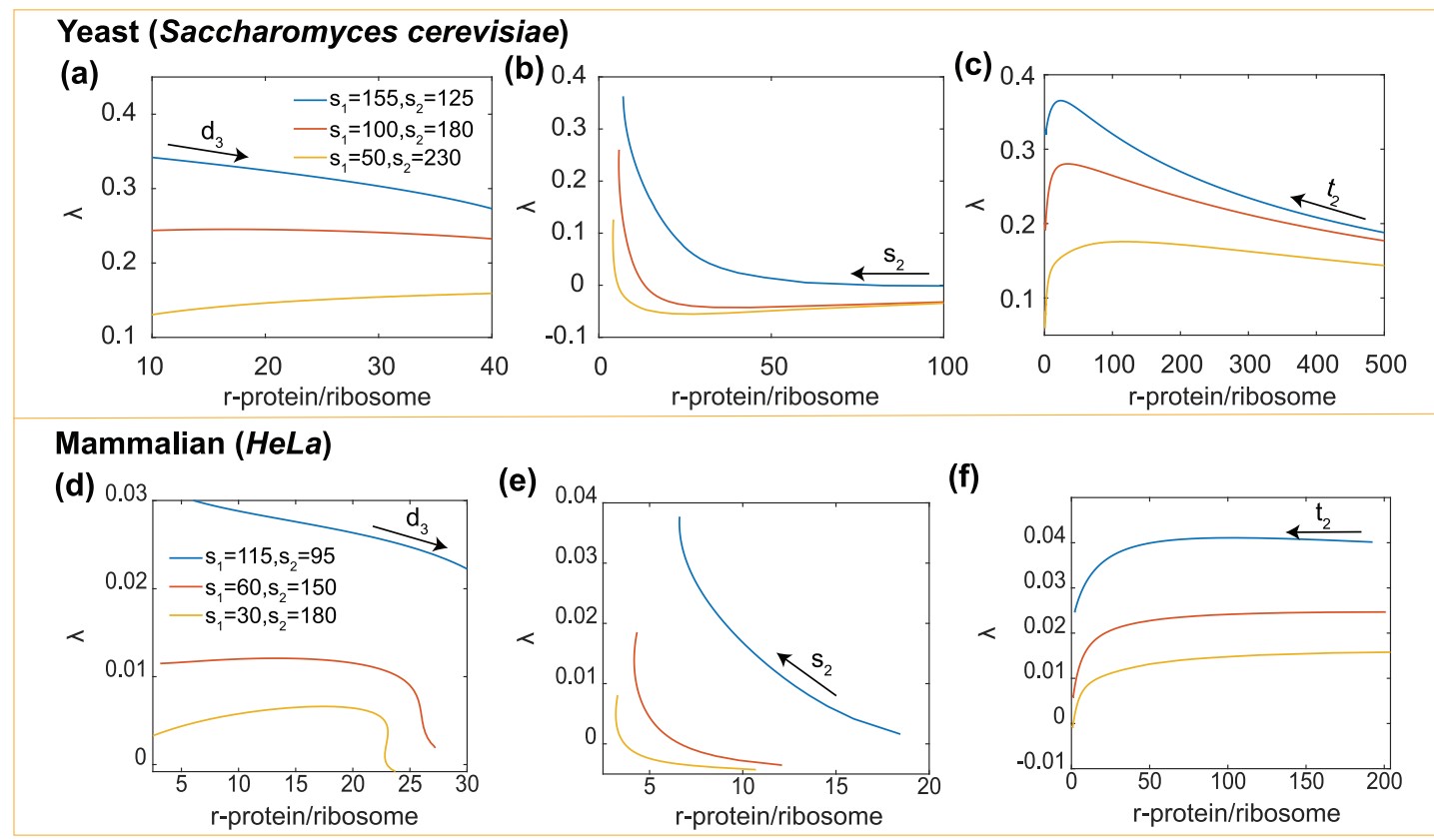

**Fig 6.** Non-monotonic effects of ribosomal protein to ribosome ratio (RP/R) on growth rate when tuning (a) disassembly coefficient $d_3$, (b) ribosome synthesis coefficient $s_3$ and (c) ribosomal protein synthesis coefficient $t_2$. (d)-(f) Corresponding results for mammalian cells. All parameters are listed in Table 2.

the growth rate decreases with this ratio. At this state, non-ribosomal proteins dominate the cell volume, and the increase of RP/R decreases the non-ribosomal protein synthesis by decreasing the ribosome number, thus decreasing the growth rate. However, when RP/R is high enough ($s_3$ is low), ribosomal proteins play a more important role in cell volume, so RP/R slightly increases growth rate by increasing the ribosomal protein number (Fig 6b and 6e). When tuning $t_2$, RP/R initially increases and then decreases the growth rate (Fig 6c and 6f). Generally speaking, the non-monotonic change of growth rate is determined by the trade-off between ribosomal protein increase (increasing macromolecule number, thus cell volume) and cytoplasmic ribosome decrease (decreasing protein synthesis). In different situations, the ribosomal proteins and cytoplasmic ribosomes play different roles, thus the RP/R-λ curve have nontrivial shapes.

## Optimal strategies for regulating cell growth and C/N ratio

Cells actively control their growth rate to adapt to changing environments. Moreover, the relative nucleus size is important in cell migration [58] and in genome regulation [59]. Therefore, it is interesting to examine how cells can control their growth rate and the C/N ratio using components of the model. We have seen that even for this simple model, many parameters can influence the cell growth rate and the C/N ratio, it is natural to ask whether the cell controls these parameters in groups, for example using canonical growth pathways. If so, does the cell have different strategies to tune the parameters under different situations? To answer these questions, we firstly normalize the parameters by their realistic values (e.g., $\bar{s}_1 = s_1/s_{10}$), and then calculate the gradients of the cell growth rate and the C/N ratio with respect to the normalized model parameters: e.g., $\partial \lambda / \partial \bar{s}_1$. The gradient vector is then normalized by its length. This can be done for the exponential growth (rich AA) and linear growth (poor AA) cases. For the quiescent case, we compute the gradient of the steady state volume with respect to parameters. Results are presented in radar charts, and the +/- sign means positive/negative derivatives (Fig 7), i.e., whether the parameter has a positive or negative influence on the quantity considered.

For both mammalian and yeast cells, when extracellular amino acid is saturating, the cell grows exponentially, and the growth rate is influenced mostly by non-ribosomal protein transport coefficients $t_4$, $t_5$ and amino acid import coefficient $t_1$ (Fig 7a and 7g)), so tuning nutrient uptake rate and reallocating protein distribution in cytoplasm and nucleus is the most effective

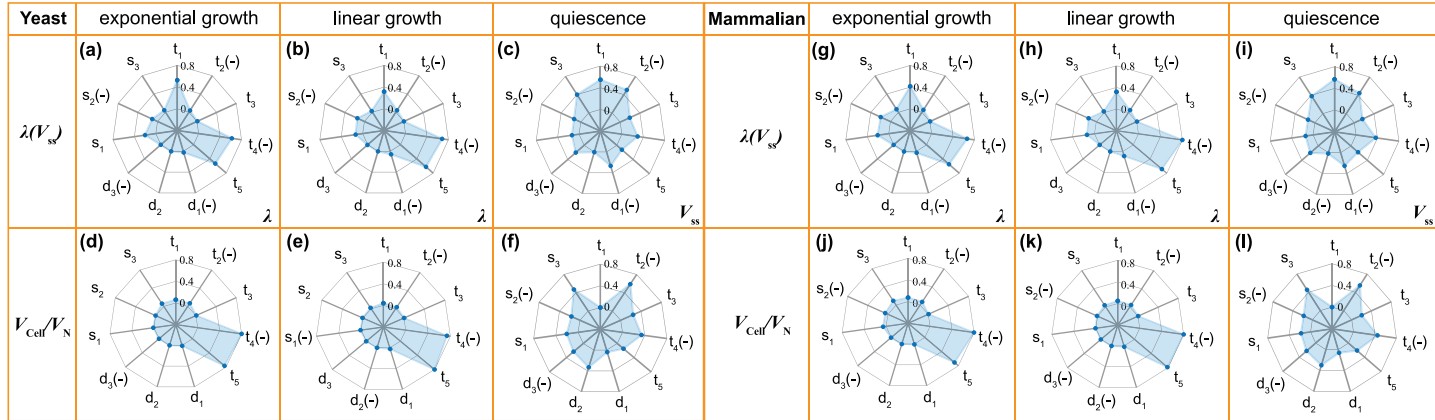

**Fig 7.** (a)-(f) Gradients of the cell growth rate and the C/N ratio with respect to the normalized model parameters for exponential, linear growth conditions, respectively. For the quiescent case, gradients of the steady state cell volume with respect to parameters are shown. Amino acid and non-ribosomal protein transport ($t_1$, $t_4$, $t_5$) are most important parameters in exponential and linear growth conditions. For the quiescent case, ribosomal protein transport ($t_2$) and ribosome synthesis ($s_3$) play most important roles. (g)-(l) Corresponding results for mammalian cell. All parameters are listed in Table 2.

way to adjust growth rate. For the C/N ratio, $t_4$, $t_5$ are the most influential parameters (Fig 7d and 7j). The AA-limited case is similar to the rich AA case, but amino acid import ($t_1$) plays a less important role in tuning cell growth (Fig 7b, 7e, 7h and 7k). In the quiescent case, the cell enters a steady state with a constant cell volume. Compared with the other two cases, ribosomal protein import ($t_2$) and ribosome synthesis ($s_3$) become the speed limiting steps in the ribosome cycle. Therefore, besides amino acid uptake ($t_1$), $t_2$ and $s_3$ are the most important parameters influencing the steady state volume and C/N ratio (Fig 7c, 7f, 7i and 7l).

In summary, in three growth conditions considered, by balancing resource uptake, allocation and utilization, the cell requires different strategies to control growth rate (steady state volume) and C/N ratio. For growing cells, besides amino acid import $t_1$, non-ribosomal protein transport coefficients $t_4$ and $t_5$ are also two of the most important parameters. These two parameters significantly influence the cytoplasmic transport protein amount, which further influence the speed of the ribosome assembly cycle. In such a case, growth resources are rich, and the determining factor of the growth is the allocation of these resources in cytoplasm and nucleus. However, in quiescent case, due to low ribosome synthesis, ribosomal protein import ($t_2$) and ribosome synthesis ($s_3$) become the speed limiting steps and have more effects in the ribosome assembly cycle and cell steady state.

### Distributions of cell volume and proteome in a growing population

With the advent of single-cell techniques, it is becoming possible to analyze the single cell volume and the single cell proteome composition. The data can be assembled to obtain a cell volume or proteome distribution for a particular cell type. Our model can be used as a starting point to obtain cell volume and proteome distributions. Mathematically, Eqs (23)–(29) are velocities of protein level and cell size increase. These velocities can be used in a stochastic model to generate cell volume and proteome distributions which has been discussed elsewhere [60, 61]. For example, a computation of protein distribution for each cell goes as follows (Fig 8a): (1) start from $N$ initial cells with model variables evenly distributed between 1 and twice

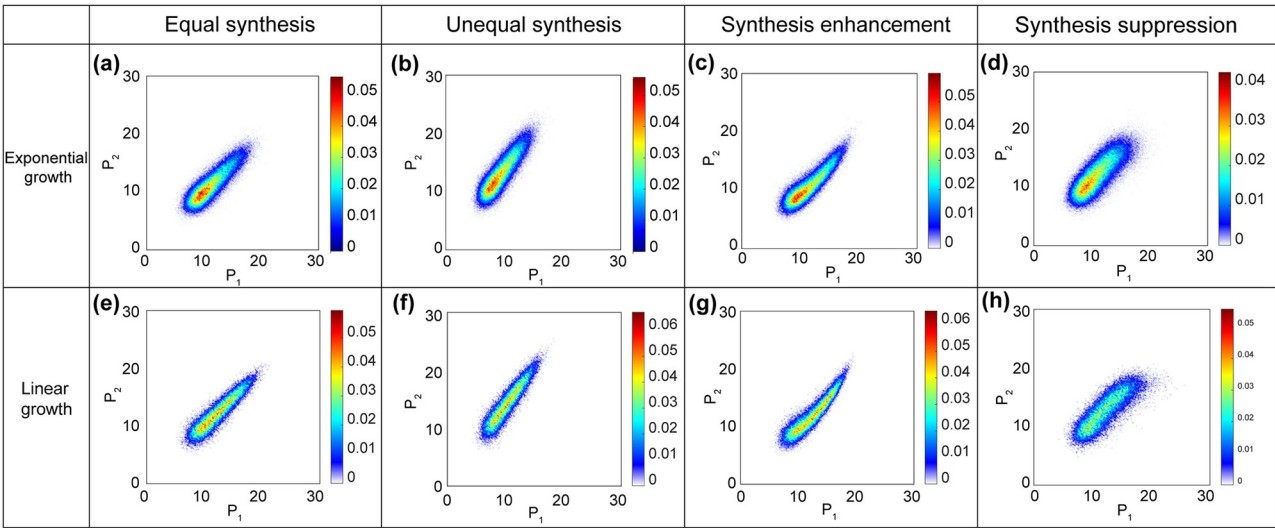

**Fig 8. Computed proteome distributions (yeast parameters) in exponential growth and linear growth.** Mammalian cell models show a similar behavior. (a) equal synthesis of $P_1$ and $P_2$ without gene regulation. The two proteins are naturally correlated because of growth through cell cycle. (b) unequal synthesis without correlation. (c) one protein enhances the synthesis of the other. (d) one protein suppresses the synthesis of the other. The distribution is not linearly correlated for synthesis enhancement and suppression cases. (e)-(h) Corresponding results for linear growth. All parameters except synthesis parameters are listed in Table 2. In the linear growth case, transport coefficient is set to be: $\bar{t}_1 = 2850h^{-1}$.

the estimated initial conditions; (2) At each time step, increment the model variables ($A_C$, $P_C$, $RP_C$, $R_C$), etc, according to governing velocities in Eqs (23)–(29) for each time step $dt = 0.02$. Then calculate the division probability density as a function of cell volume $V_C$ as [60]:

$$k(V_C) = \frac{\omega(V_C)}{1 - \int_0^{V_C} \omega(V')dV'}$$

(32)

where $\omega(V_C)$ is the division volume distribution; (3) Determine whether the cell divides during this time step: if rand$> e^{-k(V_C)dV_C}$, the cell divides symmetrically with some noise, and the volume difference between newborn cells follows a normal distribution: $\Delta V_{new}/V_{div} \sim N(0, 0.125)$ (all the constituents are divided following the same distribution) [62]; otherwise, the variables increase after the time increment. All the parameters in the model are the same as other sections. After a number of iterations, the volume and protein number distributions equilibrate to a steady distribution.

In a real cell, different proteins are translated at different rates. $s_1$ depends on the mRNA level of a particular protein, and gene regulatory interactions that may enhance or suppress the translation this particular protein. Indeed, $s_1$ should be proportional to the mRNA concentration, which can be measured in single cell RNAseq [63]. However, even without gene regulation and as long as the protein is actively translated throughout the cell cycle, we see that the protein distributions are naturally correlated, even when there are no gene regulatory interactions (Fig 8a and 8e).

To study how enhancement or suppression of expression can influence the proteome distribution, we include 2 types of non-ribosomal proteins in our model, where one protein influences the synthesis of the other. The synthesis rates are $s_{1,1}$, $s_{1,2}$, respectively. Further, we take yeast cell as an example and calculate the distributions under different synthesis situations in exponentially and linearly growing cells using the velocities from our model. In the first case, two proteins have the same synthesis rate and do not effect each other's translation. In the second case, protein $P_1$ and $P_2$ have different synthesis rates ($s_{1,1} = 65$, $s_{1,2} = 90$), but do not effect each others translation. In the third case, $P_1$ enhances the synthesis of $P_2$: $s_{1,2} = 6P_1$. In the fourth case: $P_1$ suppresses the synthesis of $P_2$: $s_{1,2} = 145 - 5P_1$. The computed protein distributions are shown in (Fig 8a–8d) for exponential growth and (Fig 8e–8h) for linear growth.

Fig 8a, 8b, 8e and 8f show that even without gene regulatory interactions, the distribution of two proteins are naturally correlated. This is because there is an overall increase of all proteins during cell growth and cell cycle. For equal synthesis rates, the slope of the correlation is 1 while if the synthesis rate are unequal, the slope can be any value. With synthesis enhancement or suppression, we can see that the distributions are more skewed and are not linearly correlated (Fig 8c, 8d, 8g and 8h). Based on Eq (2), by assuming constant ribosome concentration and equal growth rates for all the cell components, the correlation relationships between the two proteins can be approximated by: $P_2 \propto P_1^2$ and $P_2 \propto P_1 - P_1^2$ for synthesis enhancement and suppression, respectively.

Note that these computed proteome distributions will depend on the division mechanism. We have used a division probability based on Eq 32. Other mechanisms will in general give different distributions. In principle, our model can be refined to include all protein species and examine division mechanisms based on the proteome composition and cell volume.

## Discussion and conclusions

In this work, we developed a coarse-grained eukaryotic cell growth model, which includes protein and ribosome synthesis, transport across both cell surface and the nuclear envelop, protein degradation and ribosome disassembly. We simulate cell growth for yeast and mammalian

cells in three different amino acid conditions: rich-AA, poor-AA and the quiescent case. When amino acids are abundant, cells grow exponentially. However, when AA is poor, cells grow linearly due to limited AA import. When the cell enters the quiescent state, it gradually stops growing and cytoplasmic proteins are rapidly transported into nucleus, giving a low C/N ratio. In the rich-AA and quiescent cases, the C/N ratio reaches constant, but in the poor-AA case, the C/N ratio can fluctuate. Furthermore, we studied how transport, synthesis and degradation parameters influence the growth rate and the C/N ratio. Growth rate is dependent on transport across nucleus membrane. In particular, growth rate is found to be highly sensitive to non-ribosomal protein transport ($t_4$ and $t_5$), which means the cell should have tight control over these parameters. For a given total protein production rate, there exists an optimal ratio between ribosomal and non-ribosomal protein synthesis, at which the cell growth rate can achieve maximum, similar to what has been found in bacterial cells [18]. We also show that in addition to transport, protein synthesis parameters also influence the C/N ratio significantly. Ribosome disassembly coefficient $d_3$ can influence both the growth rate and the C/N ratio non-monotonically, which is due to the trade-off between disassembly-caused macromolecule number increase and cytoplasmic ribosome number decrease. Moreover, we show that the ribosomal protein to ribosome ratio RP/R also influences the growth rate non-monotonically, which could explain the size abnormality and growth rate abnormality in aneuploid cells.

It should be noted that all the kinetic parameters in the model are likely to be actively controlled in cells. Canonical growth pathways such as the AKT [64], TGF-beta [65], MTor [66] and Hippo pathways [67], and nuclear transport pathways such as the RanGTP cycle have been identified to influence cell growth and nucleoplasm-cytoplasm transport. The influence of these pathways on model parameters have yet to be studied systematically. Moreover, in different growth environments, our model predicts that the cell should utilize different strategies to achieve different growth rate or steady state volume. For exponentially or linearly growing cells, amino acid and non-ribosomal protein transport are the most important parameters, while amino acid import coefficient $t_1$ is more important for exponentially growing cells. However, for quiescent cells, ribosomal protein import and ribosome synthesis coefficients become the most important parameters. These strategies for optimal growth apply for both yeast and mammalian cells. The identified growth pathways are likely to utilize feedback mechanisms to control cell growth. This opens the possibilities of non-monotonic changes in the growth rate: for instance, time delays in the feedback mechanisms will introduce growth rate oscillations [68].

Experiments also shows considerable randomness in the cell growth rate, the cell volume and the C/N ratio, even for isogenic cells at the same point in the cell cycle [69]. This suggests that there is significant stochasticity in all the parameters in the model. The significance of this stochasticity is unclear, but could be related to how stochastic dynamical systems respond to changes in environmental conditions [70]. A system exhibiting more randomness also responses to changes faster [71]. This allows the dynamical system to rapidly adapt to new environments. Therefore, the observed cell size and growth fluctuations may be a feature of cell control system that regulate the cell physiological state.

Our model can be generalized to examine specific proteins with differing synthesis rates. The results can be compared with single cell proteomic data. By examining a model with 2 proteins, we find that the computed proteome distribution will have a trivial correlation if the proteins are synthesized independently. This is because of cell cycle effects where older cells will generally have larger number of all translated proteins. However, if the the proteins can mutually enhance or suppress their expression, additional correlations will show up in the proteome distribution. From single cell proteomics data, our model maybe used to infer gene regulatory interactions.

An interesting aspect of the model is that it predicts cell mechanical behavior can potentially couple to cell growth. There are several places where this potential coupling can occur. One, from studies of cell water and ion homeostasis, the cell volume is predicted to be proportional to the total number of macromolecules in the cell (proteins and protein complexes). The proportionality constant, however, depends on the cytoplasmic hydraulic pressure and therefore could depend on cytoskeletal contractility. Two, water and ions are generally thought to freely diffuse across the nuclear pores. However, because the nuclear envelope is mechanically connected to the cytoskeleton through LINC complexes, there may be additional mechanical forces on the nuclear envelope that balances increased hydraulic and osmotic pressures in the nucleus. Moreover, mechanical forces on the nuclear envelope may also alter nucleoplasm-cytoplasm transport rate by changing the tension in the nuclear membrane and the nuclear pore complex. Finally, direct connection between cell surface tension and cell growth pathways has been found for yeast and mammalian cells. Therefore, cell mechanical behavior can directly influence cell growth. The coupling of growth to mechanics of the cell and tissue is fundamental for our understanding of morphogenesis and this paper outlines a framework where force-dependent growth can be examined in more detail.

## Methods: Experimental measurements of cell/nucleus volume

Single cell volume and nuclear volume are measured on 500Pa polyacrylamide gels. To fabricate gels, acrylamide (3.5% w/v, Bio-Rad Laboratories Hercules, CA) and bisacrylamide (0.33% w/v, Bio-Rad Laboratories Hercules, CA) were mixed with water and a saturated solution of NHS-acrylate (N-hydroxysuccinimide ester of acrylic acid, Sigma, St. Louis, MO) in toluene (23% v/v). After mixing, the aqueous solution was removed from beneath the toluene layer. To polymerize the gel, tetramethylethylenediamine (0.3% v/v) and ammonium persulfate (0.06% w/v) were added to the acrylate mixture. The solution was allowed to polymerize for 20 minutes between two coverslips, one functionalized with glutaraldehyde and the other treated with SurfaSil. The SurfaSil-treated coverslip was removed after polymerization and the gels were rinsed twice in phosphate buffered saline (PBS). Immediately after rinsing, a 0.1 mg/ml collagen (BD Biosciences, San Jose, CA) solution in pH 8.0 HEPES buffer was added to the gels and allowed to incubate overnight at 4C. After incubation, gels were rinsed and kept in PBS before use.

For cell and nuclear volume measurements, all cells were cultured under conditions recommend by the Physical Sciences of Oncology Network [72]. Cells were cultured until passage three and then plated at 20 cells/mm2 and allowed to adhere to gels overnight under standard incubation conditions for each cell type. Cell volume was measured using a procedure adapted from previous work [73]. Briefly, immediately prior to imaging, cells were fluorescently labelled according to the manufacturer's instructions with CellTracker Green CMFDA (Invitrogen, Waltham, MA) and DRAQ5 (Cell Signaling Technology, Danvers, MA) to label cytoplasm and cell nucleus respectively. Imaging was performed on a confocal laser scanning microscope (Leica TCS SP5, Wetzlar, Germany) using a 63x, 1.2 NA water immersion objective and cells were maintained in a stage top environmental chamber during imaging. Isolated cells, that is cells separated by at least one cell diameter from neighbours, were selected for imaging. The 488 nm and 633 nm laser lines on the microscope were used to simultaneously to record signals from CMFDA for cell volume and DRAQ5 for nuclear volume. To calculate cell and nuclear volume, images were deconvolved using a precalculated point spread function, after which cell boundaries were determined using Otsu's method. All voxels within the cell boundary were included in the calculation of cell volume.

## Supporting information

**S1 Text. Supporting information, S1 Text, with additional discussions of model parameters, analytic approximations, and stability of model results.**
(PDF)

## Author Contributions

**Conceptualization:** Yufei Wu, Adrian F. Pegoraro, Paul Janmey, Sean X. Sun.

**Data curation:** Adrian F. Pegoraro, David A. Weitz, Sean X. Sun.

**Funding acquisition:** David A. Weitz, Sean X. Sun.

**Investigation:** Yufei Wu, Adrian F. Pegoraro.

**Methodology:** Yufei Wu, Adrian F. Pegoraro, Sean X. Sun.

**Project administration:** Sean X. Sun.

**Supervision:** David A. Weitz.

**Visualization:** Sean X. Sun.

**Writing – original draft:** Yufei Wu, Sean X. Sun.

**Writing – review & editing:** Yufei Wu, Paul Janmey, Sean X. Sun.

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
