## [Decision Letter · Decision Letter 0]

11 Oct 2021

Dear Dr. Sun,

Thank you very much for submitting your manuscript "The Correlation Between Cell and Nucleus Size is Explained by an Eukaryotic Cell Growth Model" for consideration at PLOS Computational Biology.

As with all papers reviewed by the journal, your manuscript was reviewed by members of the editorial board and by several independent reviewers. In light of the reviews (below this email), we would like to invite the resubmission of a significantly-revised version that takes into account the reviewers' comments.

We cannot make any decision about publication until we have seen the revised manuscript and your response to the reviewers' comments. Your revised manuscript is also likely to be sent to reviewers for further evaluation.

Sincerely,

Oleg A Igoshin

Associate Editor

PLOS Computational Biology

Jason Haugh

Deputy Editor

PLOS Computational Biology

Reviewer's Responses to Questions

**Comments to the Authors:**

Reviewer #1: The work "The Correlation Between Cell and Nucleus Size is Explained by a Eukaryotic Cell Growth Model" used a computational model to quantify factors influencing the cell volume to nuclear volume ratio (C/N ratio). While the regulatory mechanism of C/N ratio is of scientific importance and this work exhibit how simple models could explain complex biological phenomena, several major issues should be resolved before publishing this work as a formal academic paper.

Major issue:

This work heavily relies on simulation, while more quantitative insights on C/N ratio could be obtained by analytical approximations.

The major component of this work, Eq. 23-29. is in the unit of [number per cell], instead of concentration. Once they are changed in to the unit of concentration, one can work towards the steady-state solution. This could be done by dividing a "number" quantity X by their corresponding volumes V, changing into a concentration quantity x:

if dX/dt=F(X),

d(X/V)/dt=(1/V)*dX/dt-(X/V)*(1/V)*dV/dt

as the increasing speed of volume relative to the current volume, (1/V)*dV/dt, can be viewed as the growth rate g, one could obtain a differential equation of the concentration x, with an growth-induced dilution term:

d(X/V)/dt=dx/dt=(1/V)*F(X)-x*g;

for nucleus-related term, it is g_N; for cytoplasm-related term, it is g_C;

b. With the growth-induced dilution term, Eq. 23-29, when changed into the concentration format, should yield a steady-state solution about the concentration of proteins and ribosomes in the nucleus and the cytoplasm. Under this work's assumption of "cell volume is proportional to the protein and macromolecular number" (which itself needs more support from previous researches), a constant C/N ratio would be an intuitive result of "equilibrium between in/out nucleus transportation and diffusion". Then one can investigate how parameters influencing transportation and diffusion influence the relation between g_C and g_N, and the stability of C/N ratio.

2. The current manuscript lacks a clear emphasis on "which scientific questions were solved by this work". While C/N ratio is a field of interest by many researchers (which is also not yet sufficiently reviewed in this current manuscript), is the current manuscript focusing on how a constant C/N ratio can be maintained, or which biological processes may influence this ratio? What are the model results that are supported by previous observations, and what are the unexpected findings that inspire further investigation? For example, this work suggested "for the nutrient-limited cell, the C/N ratio fluctuates periodically", and " s1 and s1r have opposite effects on the C/N ratio" has that been observed in previous experiments?

3. Other than how different parameters influences C/N ratio, the format of equations should also be discussed: Which of the three processes, synthesis/transport/degradation, or subprocesses, plays essential roles in keeping the C/N ratio stable? It seems that the transportation and diffusion between the nucleus and cytoplasm are sufficient to maintain the C/N ratio stable, then what are the roles played by ribosome and protein synthesis? Is the degradation part necessary in this model?

Also, does the assumed format of equations influence the modeling results? Some assumptions in this manuscript are different from previous works: This work assumes a linear relationship between protein synthesis rate and the amino acid concentration, both in nutrient-rich and nutrient-limited conditions, which are different from the Michaelis–Menten format in Scott's 2014 work about the growth law; Also, the square relationship between protein number and its degradation rate is different from the linear degradation term in most modeling works. These differences should be supported, and whether they influence the modeling results should be discussed.

4. While this work focuses on C/N ratio in eukaryotes, the size of the nucleoid in bacterias also scales with the cell size (Gray, William T., et al. "Nucleoid size scaling and intracellular organization of translation across bacteria." Cell 177.6 (2019): 1632-1648.; Cantwell, Helena, and Paul Nurse. "Unravelling nuclear size control." Current genetics 65.6 (2019): 1281-1285). Yet, there is no distinction between the two forms of ribosomes in bacteria, as the size control mechanism proposed in this manuscript. In discussion, a comparison between the size control in eukaryotes and prokaryotes should be meaningful.

Other issues include:

Formating: Eq 22-23 extends outside of the page limit. Eq. 23-29 has no labeling.

In line 285, "ribosome disassembly and protein degradation rates in rapidly growing cells are 1/10 of the synthesis rate" is pretty arbitrary. The degradation rate can be estimated by the protein synthesis rate and the cell mass.

In Fig.4 C, the decrease of cell volume in quiescent status into near-zero value within 2 hours is not very realistic. Is that due to the assumed form of degradation?

Reviewer #2: See attachment

Reviewer #3: The review is uploaded as an attachment.

**Have the authors made all data and (if applicable) computational code underlying the findings in their manuscript fully available?**

Reviewer #1: Yes

Reviewer #2: **No: **To be useful, the simulation model should be presented as a code that can be downloaded and the parameters presented in Tables in the supporting Information or as input files for the program.

Reviewer #3: None

PLOS authors have the option to publish the peer review history of their article (what does this mean?). If published, this will include your full peer review and any attached files.

Reviewer #1: No

Reviewer #2: **Yes: **Zaida Luthey-Schulten

Reviewer #3: No
---

## [Decision Letter · Decision Letter 1]

20 Dec 2021

Dear Dr. Sun,

Thank you very much for submitting your manuscript "The Correlation Between Cell and Nucleus Size is Explained by an Eukaryotic Cell Growth Model" for consideration at PLOS Computational Biology. As with all papers reviewed by the journal, your manuscript was reviewed by members of the editorial board and by several independent reviewers. The reviewers appreciated the attention to an important topic. Based on the reviews, we are likely to accept this manuscript for publication, providing that you modify the manuscript according to the review recommendations.

Sincerely,

Oleg A Igoshin

Associate Editor

PLOS Computational Biology

Jason Haugh

Deputy Editor

PLOS Computational Biology

[LINK]

Reviewer's Responses to Questions

**Comments to the Authors:**

Reviewer #1: In the revised version of "The Correlation Between Cell and Nucleus Size is Explained by an Eukaryotic Cell Growth Model", most of the concerns from reviewers were appropriately addressed. Current, I think it is a nice scientific story that aims at solving fundamental questions in cell biology by computational methods with clear logic. However, I still have some concerns about the stability of the C/N ratio.

In the reply for comment 1, the author states that "we can express growth rate as _ = _ = ", which would already lead to a proportional C/N volume, as a result of the exponential growth with the same exponents. From the current concentration equations the authors listed in the reply, can the Jacobian be calculated to prove the stability of the fixed point solution?

In the line "689", there is an "?" that should not be there.

Reviewer #3: The authors have satisfactorily responded to most of my concerns and suggestions. The manuscript reads better now and is more informative. However, we seem to disagree on how to model cellular volume. The authors seem convinced that the cell is a dilute media whose volume is solely determined by its osmolarity, i.e., the cellular volume is proportional to the number of molecules of proteins and ribosomes and is independent of their sizes. This assumption contradicts the modern lore of the cell being a crowded environment and the fact that ribosomes and DNA are significantly large molecules. I should mention that I am mainly aware of the literature on Prokaryotes. I do not know if the authors' assumptions are indeed accurate for Eukaryotes. Also, as there is no experimental evidence of the predicted dependence of cell/nucleus volume on the various parameters studied, it is difficult to assess the correctness of the model. However, I will still recommend acceptance. As the saying goes - all models are wrong, but some are useful. Hopefully, this work will encourage further research on this topic. It will be interesting to see if different modelling approaches predict different outcomes and if experimental observations can differentiate between them.

**Have the authors made all data and (if applicable) computational code underlying the findings in their manuscript fully available?**

Reviewer #1: Yes

Reviewer #3: Yes

PLOS authors have the option to publish the peer review history of their article (what does this mean?). If published, this will include your full peer review and any attached files.

Reviewer #1: No

Reviewer #3: No

Figure Files:

Data Requirements:

Reproducibility:

References:

---

## [Editor Report · Decision Letter 2]

12 Jan 2022

Dear Dr. Sun,

We are pleased to inform you that your manuscript 'The Correlation Between Cell and Nucleus Size is Explained by an Eukaryotic Cell Growth Model' has been provisionally accepted for publication in PLOS Computational Biology.

Best regards,

Oleg A Igoshin

Associate Editor

PLOS Computational Biology

Jason Haugh

Deputy Editor

PLOS Computational Biology

---

## [Editor Report · Acceptance letter]

11 Feb 2022

PCOMPBIOL-D-21-01529R2 

The Correlation Between Cell and Nucleus Size is Explained by an Eukaryotic Cell Growth Model

Dear Dr Sun,

I am pleased to inform you that your manuscript has been formally accepted for publication in PLOS Computational Biology. Your manuscript is now with our production department and you will be notified of the publication date in due course.

With kind regards,

Zsanett Szabo
